# A Tale of LLMs and Induced Small Proxies: Scalable Agents for Knowledge Mining

## Abstract

At the core of Deep Research is knowledge mining, the task of extracting structured information from massive unstructured text in response to user instructions. Large language models (LLMs) excel at interpreting such instructions but are prohibitively expensive to deploy at scale, while traditional pipelines of classifiers and extractors remain efficient yet brittle and unable to generalize to new tasks. We introduce Falconer[1], a collaborative framework that combines the agentic reasoning of LLMs with lightweight proxy models for scalable knowledge mining. In Falconer, LLMs act *as planners*, decomposing user instructions into executable pipelines, and *as annotators*, generating supervision to train small proxies. The framework unifies classification and extraction into two atomic operations, `get_label` and `get_span`, enabling a single instruction-following model to replace multiple task-specific components. To evaluate the consistency between proxy models incubated by Falconer and annotations provided by humans and large models, we construct new benchmarks covering both planning and end-to-end execution. Experiments show that Falconer closely matches state-of-the-art LLMs in instruction-following accuracy while reducing inference cost by up to 90% and accelerating large-scale knowledge mining by more than 20x, offering an efficient and scalable foundation for Deep Research.

## 1 Introduction

Knowledge mining tasks (Xu et al.; Boylan et al., 2025; Wang et al., 2025; Ma et al., 2024; Walker et al., 2006a) require processing massive corpora, extracting structured information, and generating annotations at scale (Ding et al., 2021; Tedeschi & Navigli, 2022; Li et al., 2023; Bogdanov et al., 2024; Peng et al., 2024). Characterized by the need to faithfully follow user instructions, these tasks often involve millions of records, such as parsing customer reviews, analyzing biomedical literature, or summarizing large collections of technical documents. The sheer scale makes efficiency critical: any system must deliver accurate results while handling high throughput at low cost. Large language models (LLMs) provide strong instruction-following capabilities (OpenAI, 2025; Anthropic, 2025; Comanici et al., 2025) and achieve high accuracy on such tasks (Agrawal et al., 2022; Wang et al., 2023b; Xu et al., 2024a). However, using LLMs directly as the executors of knowledge mining pipelines is computationally prohibitive. Each API call incurs substantial latency and cost, and iterating over millions of records quickly becomes infeasible. Thus, while LLMs are powerful, they are simultaneously too expensive and overqualified for large-scale knowledge mining. At the other extreme, traditional knowledge mining systems rely on chaining classifiers and extractors (e.g., named entity recognition models) to achieve efficiency. However, these systems lack the instruction-following ability of LLMs, forcing developers to manually construct rigid, task-specific pipelines. For instance, to carry out the instruction *Extract all laptop prices from positive Amazon reviews*(Figure 1), one must hand-engineer a sequence of modules. First, a classifier must be trained to determine whether a given review is a positive review about laptops. Next, an extractor must be trained to identify and extract the price information from those filtered reviews.

---

[1]A falconer is one who trains and guides falcons in the hunt, and we adopt this name because our framework similarly uses a central LLM to "train and direct" lightweight proxy models that swiftly pursue labels and spans across massive corpora.

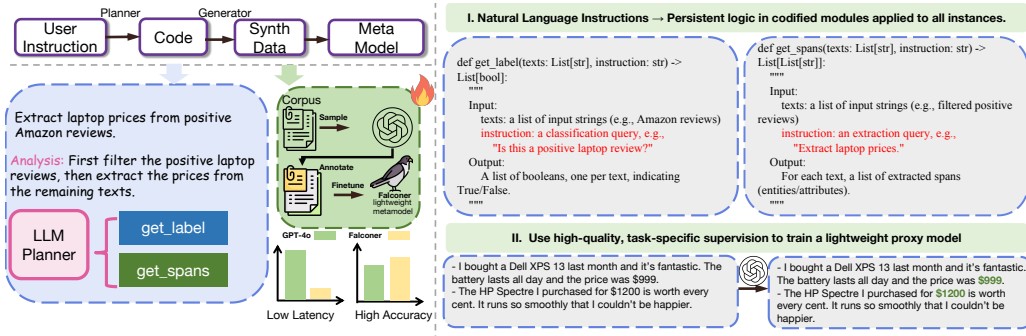

Figure 1: Falconer decomposes the instruction *Extract all laptop prices from positive Amazon reviews* into `get_label` and `get_spans`, generates supervision for training the competent proxy, and executes these primitives efficiently with small-model inference. On the right, we show how Falconer instantiates the subtasks: first classifying reviews as positive laptop reviews, then extracting the corresponding price spans. This design enables Falconer to combine the instruction-following ability of LLMs with the efficiency of small models.

To overcome this limitation, we replace hand-crafted pipelines with the agentic behavior of LLMs. LLMs serve two complementary roles. As **planners**, they decompose natural language instructions into structured subtasks (e.g. first classify whether a review is a positive laptop review, then extract its price), removing the need for manual pipeline design. As **annotators**, they provide high-quality supervision for training lightweight proxies, enabling small models to execute the subtasks efficiently at scale. In Falconer, diverse SLMs are unified into two primitive operations: `get_label(text, instruction)`, which performs classification, and `get_span(text, instruction)`, which extracts relevant spans. These two functions act as the atomic building blocks for knowledge mining pipelines. For example, to process the previous instruction, the pipeline first calls `get_label($review, 'Is this a positive laptop review?')` to filter reviews, and then applies `get_span($review, 'Extract the price')` to identify price mentions. More complex tasks, such as relation extraction or multi-entity queries, can be similarly expressed as sequences of these two primitives.

| Methods | Pipeline Design | Instruction Following | Executor | Modeling Paradigm | Efficiency | Scalability Corpus-level | Task-level |
|---|---|---|---|---|---|---|---|
| Traditional Pipeline | Manual chaining | ✗ | Separate classifiers + Extractors | Schema-based | High | ✓ | ✓ |
| Direct LLM Executor (OpenAI, 2024) | None (end-to-end) | ✓ | Large LLM API | Generative | Low | ✗ | ✓ |
| RoBERTa Baseline (Liu et al., 2019) | Manual schema-based | ✗ | Multiple RoBERTa models | Discriminative | Medium | ✓ | ✗ |
| MetaIE (Peng et al., 2024) | Synthetic schema | Partial | Distilled proxy model | Hybrid | Medium | ✓ | ✓ |
| Cuckoo (Peng et al., 2025) | Instruction-tuned IE | ✓ | Single lightweight proxy | Extraction + Classification | High | ✓ | ✓ |
| **Falconer (Ours)** | LLM planner + Annotator | ✓✓ | Unified competent proxy | Planner + Proxy | **High** | ✓ | ✓ |

Table 1: A comparison of Falconer with traditional pipelines, direct LLM executors, and lightweight baselines. Falconer uniquely combines LLM planning and annotation with a unified competent proxy, achieving both instruction-following flexibility and efficiency at corpus scale.

This design incubates and integrates all pipeline components in a unified manner, rather than engineering them separately. Whereas traditional systems required distinct models for each step. For instance, executing the previous instruction, a traditional pipeline needs at least two models: first a classifier (e.g., RoBERTa (Liu et al., 2019)) to detect positive laptop reviews, and a span extractor to identify price mentions. Such components demand separate training and maintenance, which increases cost and compounds errors across the pipeline. Moreover, these models cannot directly interpret instructions: labels such as "positive review" or "price" must be predefined. Instead, Falconer leverages **Cuckoo** (Peng et al., 2025), a high-capacity instruction-following proxy trained under the NTE paradigm. Cuckoo unifies classification and extraction within a single lightweight model, abstracted as `get_label(text, instruction)` and `get_span(text, instruction)`. Crucially, it is instruction-aware: it can directly follow prompts such as *Is this a positive laptop review?* or *Extract the price*, without relying on fixed label sets or schema-specific engineering. This allows Falconer to replace brittle, hand-crafted pipelines with a single adaptive model that retains both the efficiency of small models and the flexibility of LLM-style instruction following.

Due to the absence of instruction-following benchmarks for knowledge mining, we design new evaluations that test both planning ability and end-to-end performance. These benchmarks assess the consistency of Falconer proxies with annotations from humans and large models. Results reveal that

while LLMs excel as planners, their scalability is inherently limited. By contrast, Falconer achieves end-to-end performance that closely tracks state-of-the-art LLMs, establishing it as an efficient and practical alternative to purely LLM-based knowledge mining pipelines.

In summary, our main contributions are threefold:

- We propose Falconer, a framework where LLMs serve as planners and annotators, decomposing natural language instructions into pipelines and generating supervision for lightweight proxies.
- We introduce an **instruction-following proxy** that unifies classification and extraction into two atomic operations (`get_label`, `get_span`), enabling a single small model to replace multiple task-specific components.
- We construct new **instruction-following benchmarks** for knowledge mining, evaluating both planning and end-to-end execution. Experiments show that Falconer closely tracks state-of-the-art LLMs while cutting inference cost by up to 90% and accelerating large-scale processing by over 20×.

## 2  RELATED WORKS

**Information Extraction**   Information extraction (IE) is one of the most fundamental applications in knowledge mining. IE systems take the user's requirement (e.g., defined by a label text, a question, or an instruction) and extract spans of several tokens from input texts. IE encompasses a wide range of task formulations with different level of difficulties, which varies from simple structure entity and relation extraction such as named entity recognition (Sang & De Meulder, 2003), relation extraction (Carreras & Màrquez, 2005) , and event extraction (Walker et al., 2006b), to more difficult tasks such as abstratc entity extraction (Pontiki et al., 2016; Xu et al., 2020).

**LLM Agents**   Recent work leverages the advanced reasoning and comprehension abilities of large language models (LLMs) to tackle diverse downstream tasks (Besta et al., 2024; Yao et al., 2023a; Shinn et al., 2023). For complex scenarios, LLMs have been framed as autonomous agents that interact with environments (Chen et al., 2023; Yao et al., 2023b; Lu et al., 2023), employ external tools (Wu et al., 2024; Zong et al., 2024; Peng et al., 2023; Durante et al., 2024), and accumulate experiential knowledge (Fu et al., 2024; Zhao et al., 2024). A representative example is ReAct (Yao et al., 2023b), which tightly integrates reasoning and action by alternating between intermediate reasoning and external operations such as information retrieval.

**LLM Agents for Retrieval**   LLM agents have been applied to Information Retrieval (IR) through pretraining, reranking, and prompting (Zhuang et al., 2023; Shen et al., 2023; Wang et al., 2023a). As retrievers directly impact downstream tasks such as retrieval-augmented generation (Lewis et al., 2020) and knowledge-intensive QA, domain-specific agents like EHRAgent (Shi et al., 2024) have been developed to incorporate structured tool-use planning process and an interactive coding mechanism. Nevertheless, existing approaches largely depend on heuristic prompts or few-shot examples, providing limited guidance for effective retrieval strategies and tool-assisted actions.

## 3  FALCONER

Our framework is mainly composed of 3 components: planner, generator and a compact proxy metamodel that nonetheless exhibits robust performance across diverse tasks. An overview of the framework is provided in Figure 1. Our framework takes a task prompt and output specification, uses a planner to generate execution code, and then leverages the generator and metamodel to produce a fine-tuned model for execution. This yields a fully automated pipeline where users simply provide text and obtain high-quality outputs, achieving a twenty-fold speedup and a 90% cost reduction compared with GPT-4o, while maintaining strong performance.

### 3.1  PRELIMINARIES

INSTRUCTION-FOLLOWING PROXY MODEL: CUCKOO

The Next Token Prediction (NTP) paradigm equips LLMs with broad semantic knowledge and impressive instruction-following ability but lacks explicit token-level supervision for information

extraction (IE). To simultaneously attain robust instruction-following capabilities and fine-grained token-level supervision, Cuckoo (Peng et al., 2025) proposes the Next Tokens Extraction (NTE) paradigm, which automatically converts repeated spans in raw corpora into BIO-labeled data, turning unannotated text into large-scale IE supervision. Cuckoo leverages both pre-training and post-training resources from LLMs to build powerful NTE-based information extraction models:

- **Pre-training**: Conducted on large-scale C4 (CommonCrawl) dataset (Raffel et al., 2020). NTE automatically generates BIO labels for repeated spans, enabling the model to learn general-purpose extraction abilities without manual annotation.
- **Post-training**: Conducted on Tülu 3 (Lambert et al., 2024), a *diverse* and *high-quality* publicly available dataset. Unlike pre-training, only NTE labels relevant to user instructions are retained, equipping the model with strong instruction-following capabilities.

Under the few-shot setting, Cuckoo and its variant achieve stronger performance than existing pretrained IE models. We adopt Super Rainbow Cuckoo[2], a variant further trained on additional datasets, as our metamodel due to its superior extraction, QA, and classification abilities, as well as its strong instruction-following capability for versatile downstream tasks.

## 3.2 Cuckoo for Text Classification

The original Cuckoo model is speciliazed in Basic IE (Information Extraction) tasks such as entity extraction and relation extraction, Query-based IE and Instruction-Following IE (Peng et al., 2025). Leveraging Cuckoo's instruction-following capability, we could further extend its applicability to text classification tasks through the design of tailored prompt templates. Specifically, text classification can be reformulated as a natural language inference (NLI) problem, where the goal is to determine the relationship between a given sentence and a candidate label—namely, whether the sentence entails the label. To this end, we construct an instruction-based prompt template for classification and fine-tune the Super Rainbow Cuckoo model on the datasets introduced in Laurer et al. (2023), yielding the metamodel employed in our experiments. Further details of fine-tuning are provided in Appendix A.

## 3.3 Planning

The planner is the core of Falconer, translating natural language requirements into executable pipelines by codifying instructions into atomic operations and explicit control flows. For a knowledge mining objective, it decomposes the input into subtasks (e.g., classification, span extraction), each bound to a tool interface such as `get_label` or `get_span`. These are then assembled into a deterministic control flow, ensuring explicit execution without reliance on implicit reasoning. Sample code is shown in Appendix B.

Crucially, the planner does not merely synthesize runnable code but codifies the logical dependencies among subtasks. For example, in a multi-entity extraction scenario, *Retrieve all talks about both health and brain, then extract their lecturers*, the planner constructs a sequential program where the input texts are first filtered using two classification heads for "health" and "brain," then conditionally passed into a span extractor to identify lecturer names. This approach integrates boolean logic, ordered execution, and parameterized prompt templates into a unified representation, ensuring that downstream behavior is both interpretable and reusable across tasks.

By explicitly codifying instructions into executable task pipelines, Falconer achieves two key benefits. First, the structured representation allows the planner to generalize across diverse task formulations, including multi-label classification and multi-entity extraction. Second, codification improves transparency: every decision taken by the system can be traced back to a deterministic plan, bridging the gap between user intent and model actions.

Table 2 compares the planning abilities of different models. We observe that GPT-4.1 achieves high accuracy across diverse tasks, making it a strong candidate for our planner. However, performance drops on complex tasks, which we define as multi-step tasks that require intermediate execution results rather than a single fixed string (e.g., first extracting a lecturer's name, then identifying that lecturer's profession). To further probe model limits, we include a set of miscellaneous tasks specifically

---

[2]https://huggingface.co/KomeijiForce/Cuckoo-C4-Super-Rainbow

designed to stress-test state-of-the-art LLMs under such challenging scenarios. While models struggle in these cases, their accuracy improves substantially with in-context learning (ICL), underscoring both the difficulty of complex tasks and the effectiveness of our framework in decomposing knowledge mining objectives into well-structured subtasks.

| Method | Basic | Query-Based | Multi-Entity | Misc. | Misc. w/ In-Context Learning |
|---|---|---|---|---|---|
| Falconer w/ GPT-4.1 | **0.96** | **1.00** | **1.00** | **0.21** | **0.96** |
| Falconer w/ GPT-4o | 0.63 | 0.78 | 1.00 | 0.19 | 0.84 |
| Falconer w/ Claude 3.7 Sonet | 0.78 | 0.80 | 0.98 | 0.19 | 0.92 |
| Falconer w/ GPT-4o-mini | 0.50 | 0.19 | 0.30 | 0.00 | 0.42 |

Table 2: Planning correctness score with different LLM as Planner

## 3.4 GENERATOR

One major challenges in adapting a lightweight metamodel to diverse knowledge mining tasks lies in acquiring high-quality, task-specific supervision without incurring prohibitive costs. In Falconer, we address this challenge by introducing a generator, a component designed to bridge the gap between raw corpus data and the specialized capabilities required by the metamodel. Unlike synthetic data fully produced by large language models, which often diverges from the target distribution, the generator leverages the underlying structure of knowledge mining scenarios to produce realistic and task-aligned supervision.

The generator operates in three stages. First, around five percent of the entire corpus is sampled to capture the authentic distribution of the domain, which is detailed in Appendix C. Second, a powerful large language model (e.g., GPT-4.1) annotates these samples according to the planner's codified task descriptions, covering subtasks such as entity extraction, classification, and relation detection. Importantly, the generator enriches naturally occurring data with high-quality labels rather than fabricating artificial inputs, ensuring statistical fidelity to the corpus. Finally, the annotated samples are used to fine-tune the metamodel, enabling it to acquire task-specific knowledge while maintaining its efficiency advantages over large models. A summary of performance gains is provided in Table 3.

In subsequent experiments, this approach demonstrates high efficiency, achieving performance comparable to or even surpassing state-of-the-art large language models while using only 5% of the original corpus. Crucially, the generator's success hinges on access to high-quality supervision, which can be readily extended to alternative sources such as carefully curated human annotations.

## 3.5 METAMODEL: LIGHTWEIGHT YET CAPABLE PROXY

In Falconer, the metamodel serves as the central execution engine, acting as a lightweight proxy for large language models in downstream knowledge mining tasks. Instead of relying on general-purpose LLMs for every request, we adopt Cuckoo (Peng et al., 2025), which has similar parameters as Liu et al. (2019), to strikes a balance between parameter efficiency and capability, enabling Falconer to achieve the best of both worlds: near-LLM performance with dramatically reduced inference cost.

Moreover, Falconer's modular architecture leverages Cuckoo not as a monolithic generalist, but as a specialized executor within a planner-driven pipeline. The planner codifies user intents into explicit, interpretable subtasks; the metamodel then executes these subtasks with high efficiency. This separation enables Falconer to exploit the SLM-first paradigm advocated by recent research (Belcak et al., 2025).

Empirically, this design achieves substantial gains in both efficiency and scalability. Cuckoo requires up to 20× fewer FLOPs and 1000× less memory than GPT-class models, while maintaining competitive accuracy on instruction-following and span-extraction benchmarks relevant to knowledge mining. This efficiency enables Falconer to operate cost-effectively across massive corpora, supporting real-time inference even in resource-constrained environments.

## 4 EXPERIMENTS

We evaluate Falconer on a broad spectrum of knowledge mining tasks to demonstrate that a lightweight metamodel, when coupled with our planner–generator–executor framework, can achieve performance

comparable to state-of-the-art LLMs while being significantly more efficient. Our experiments are designed to answer two central questions:

- whether these metamodels maintain high alignment with human annotations on labeled datasets and
- whether Falconer can generate metamodels that faithfully approximate the behavior of large models(its annotator) on unlabeled corpora

All experiments reported in this section were conducted using a metamodel fine-tuned on 5% of the original corpus annotated by an LLM, unless otherwise specified. Model performance is evaluated using the word-level F1 score.

## 4.1 LABELED DATASET

| Metamodel | Dataset | 64 Samples | 512 Samples | GPT-4o |
|---|---|---|---|---|
| Cuckoo | Fabrication | 0.20 | 0.32 | **0.38** |
| RoBERTa-Large | Fabrication | 0.00 | 0.00 | 0.38 |
| Cuckoo | Biology | 0.42 | **0.45** | 0.27 |
| RoBERTa-Large | Biology | 0.00 | 0.41 | 0.27 |
| Cuckoo | Twitter | 0.19 | **0.43** | 0.35 |
| RoBERTa-Large | Twitter | 0.00 | 0.38 | 0.35 |
| Cuckoo | Wiki | 0.03 | **0.68** | 0.53 |
| RoBERTa-Large | Wiki | 0.00 | 0.60 | 0.53 |
| Cuckoo | Vehicle | 0.42 | 0.75 | **0.76** |
| RoBERTa-Large | Vehicle | 0.00 | 0.66 | 0.76 |

Table 3: Results on NER Datasets with Ground Truth labels

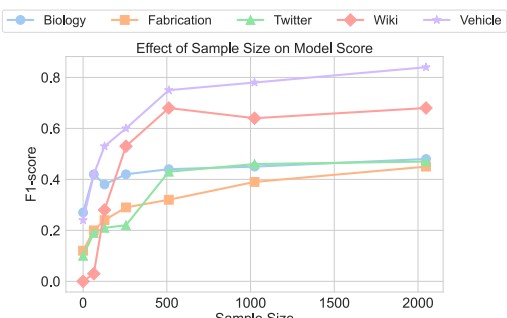

Figure 2: Model Performance under different sample size

This set of experiments is primarily intended to assess the consistency between the metamodel and human annotations, as well as to benchmark the performance of the metamodel against that of contemporary large language models. Furthermore, we utilized several widely adopted Named Entity Recognition (NER) datasets, including **FabNER**, **Broad Twitter**, **BC2GM**, **AnatEM**, **WikiNER**, and **FindVehicle**. These datasets were combined to construct a new benchmark, which was subsequently employed to assess the metamodel's performance across diverse groups of tasks. For particularly large datasets, such as WikiNER, we randomly sampled a subset to the mixed dataset. Meanwhile, to more explicitly illustrate the adaptability of the metamodel to downstream tasks, we present experimental results obtained by fine-tuning the metamodel with varying amounts of training data, ranging from 64 to 2048 samples. It is worth noting that even the largest setting of 2048 samples corresponds to only 5% of the original corpus. The main results are shown in Table 3 and detailed results are plotted in Figure 2.

From the experimental results, we observe a consistent improvement in test performance as the sample size increases. Notably, the model fine-tuned with 2048 samples **outperforms GPT-4o across all task categories**, providing strong evidence of its substantial adaptability to knowledge mining tasks. Meanwhile, the rate of performance gains is closely tied to the quality of annotations generated by the large model. When the annotations are of high quality, the metamodel tends to achieve performance saturation more rapidly, as illustrated by the experiments on WikiNER. Conversely, in tasks where the large model produces suboptimal annotations, the performance of the metamodel improves more gradually, thereby reflecting the core principle of co-evolution between the metamodel and large models (Peng et al., 2025).

## 4.2 UNLABELED DATASET EVALUATION

To evaluate the effectiveness of Falconer in generating reliable proxy metamodels, we measure the consistency scores between the metamodel and GPT-4o across three large-scale unlabeled corpora, **TED Talk Summary**, **Steam Game Description**, **and Text Message**. We design a diverse set of knowledge mining tasks spanning three categories: **basic tasks** involving entity recognition and

| | Model | Dataset | Basic Task | | | | Query-based Task | | | Multi-entity Task | | | |
|---|---|---|---|---|---|---|---|---|---|---|---|---|---|
| | | | Task 1 | Task 2 | Task 3 | Average | Task 1 | Task 2 | Average | Task 1 | Task 2 | Task 3 | Average |
| **0-shot** | Cuckoo | TED | 0.489 | 0.654 | 0.514 | 0.552 | 0.383 | 0.371 | 0.377 | 0.395 | 0.607 | 0.497 | 0.500 |
| | Cuckoo | Steam Game | 0.501 | 0.683 | 0.535 | 0.573 | 0.374 | 0.350 | 0.362 | 0.451 | 0.524 | 0.468 | 0.481 |
| | Cuckoo | Text Message | 0.584 | 0.694 | 0.585 | 0.621 | 0.418 | 0.392 | 0.405 | 0.564 | 0.583 | 0.530 | 0.559 |
| **Few-shot** | Cuckoo | TED | 0.658 | 0.758 | 0.683 | **0.699** | 0.532 | 0.557 | **0.545** | 0.644 | 0.692 | 0.661 | **0.666** |
| | Roberta-Large | TED | 0.552 | 0.587 | 0.553 | 0.564 | 0.446 | 0.511 | 0.479 | 0.517 | 0.566 | 0.531 | 0.538 |
| | Cuckoo | Steam Game | 0.672 | 0.783 | 0.675 | **0.710** | 0.569 | 0.587 | **0.578** | 0.673 | 0.719 | 0.684 | **0.692** |
| | Roberta-Large | Steam Game | 0.509 | 0.525 | 0.517 | 0.517 | 0.434 | 0.452 | 0.443 | 0.588 | 0.383 | 0.564 | 0.512 |
| | Cuckoo | Text Message | 0.703 | 0.806 | 0.731 | **0.747** | 0.590 | 0.614 | **0.602** | 0.709 | 0.726 | 0.734 | **0.723** |
| | Roberta-Large | Text Message | 0.553 | 0.621 | 0.590 | 0.588 | 0.496 | 0.518 | 0.507 | 0.548 | 0.570 | 0.574 | 0.564 |

Table 4: Results from various proposed tasks on 3 datasets with subtasks

simple classification, **query-based tasks** requiring sentence-level semantic understanding, and **multi-label/multi-entity task**s that demand compositional reasoning. Please refer to the complete list of tasks provided in Appendix E

**Basic Task**    This category benchmarks the fundamental capacity of models to discern labels, entities, and relations. We construct a suite of tasks that closely approximate real-world knowledge mining settings, exemplified by sample 1 and 2 in Appendix D. The task set spans elementary classification, entity and relation extraction, as well as composite formulations integrating both. For pairwise relation extraction tasks, we further stipulate that one entity participating in the relation is pre-specified, thereby isolating the model's ability to infer the remaining relational structure. As shown in Table 4, the tasks categorized as Basic Task demonstrate that, after fine-tuning, the metamodel consistently achieves high agreement with the large model.

**Query-Based Task**    This category of tasks focuses on assessing the model's ability to capture more complex sentence-level semantics, as exemplified by sample 3 and 4 in Appendix D. Illustrated in Table 4, the corresponding tasks are represented by Query-based Task. With appropriate fine-tuning, the metamodel demonstrates competitive performance on complex tasks. It is worth noting that the untuned metamodel exhibits the weakest performance in this category; however, fine-tuning yields substantial improvements. For instance, given the task prompt "retrieve all texts that are primarily about medicine, and extract what the lecturer will talk about", the initial metamodel achieves an F1 score of only 0.23 when compared against GPT-4o as the reference. After fine-tuning with only a small fraction of the annotated corpus, its F1 score increases to 0.56. These results highlight the model's strong capacity to adapt effectively to downstream tasks.

**Multi-entity Task**    This category of tasks extends metamodel evaluation to multi-label and multi-entity scenarios (sample 5 in Appendix D). Prior work highlights the limitations of large language models in multi-label classification (Ma et al., 2025; Xu et al., 2024b). In contrast, our framework employs the planner to decompose such tasks into sequential subtasks, whose outputs are aggregated to form the final result. For instance, the query "retrieve all speeches concerning both health and the brain" is decomposed into two classification subtasks—health-related and brain-related—whose results are combined via Boolean logic. This structured decomposition enables logically consistent and accurate performance in multi-label classification and multi-entity extraction.

The experimental results for Multi-entity Task, as reported in Table 4, indicate that the adapted metamodel demonstrates strong proficiency in handling multi-entity tasks, achieving performance that is competitive with, and in some cases surpasses, results obtained through multi-turn prompting augmented with human annotations.

## 5    ANALYSIS

### 5.1    CONTINUAL INTEGRATION ANALYSIS

In Table 3, fine-tuning was restarted from a fresh base model for each task. In practice, however, continual learning is equally important, as models are expected to sustain performance across sequential tasks while retaining competence from earlier ones. To evaluate this ability, we reformulated the setup into a sequence of five tasks, where each task used the model fine-tuned on its predecessor as the base.

We report the results in Figure 3, averaging over subtasks when applicable. The figure demonstrates the metamodel's performance under sequential fine-tuning and evaluation on consecutive tasks.

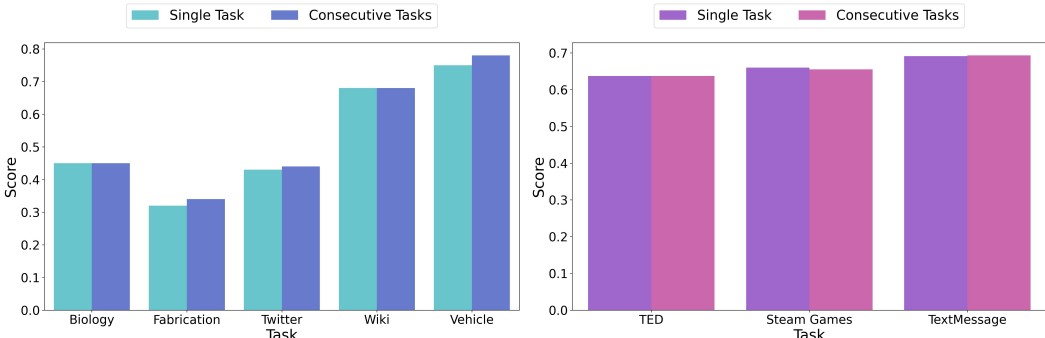

Figure 3: Performance on labeled dataset of Single Task w/ new metamodel and Consecutive Task w/ metamodel from previous task. Performance on unlabeled dataset of Single Task w/ new metamodel and Consecutive Task w/ metamodel from previous task

We observe that models undergoing multiple rounds of fine-tuning on sequential tasks maintain capabilities comparable to those fine-tuned directly from the base model. Overall, our evaluation highlights the metamodel's continual integration ability, demonstrating its effectiveness in sustaining high performance across a broad spectrum of real-world tasks. Moreover, the results validate that the proposed framework substantially alleviates the deployment overhead associated with adapting models to diverse tasks.

## 5.2 EFFICIENCY ANALYSIS

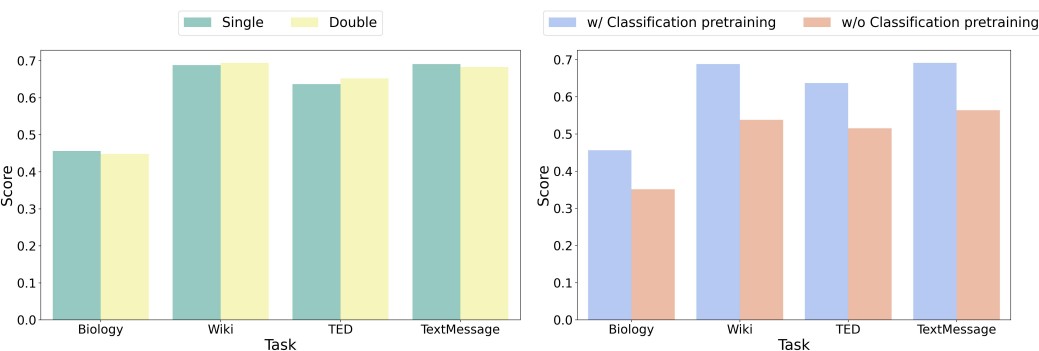

Figure 4: Performance of different number of metamodel for different task type. Performance of different pretraining strategy

In this section, we further highlight the efficiency and performance advantages of our framework. While prior experiments benchmarked RoBERTa-large against multiple baselines, its lack of inherent instruction-following ability required training two task-specific variants for classification and extraction. By contrast, our framework enables the incubation of a single metamodel that leverages instruction-following to generalize across heterogeneous tasks. To validate this, we additionally trained two separate metamodels—one for classification and one for extraction—on the same benchmarks. As shown in left panel of Figure 4, their performance is nearly indistinguishable from that of a unified model, underscoring that a single metamodel can achieve state-of-the-art performance across task types while significantly reducing deployment overhead.

Meanwhile, to further substantiate the metamodel's capacity for continual generalization across novel tasks, we additionally evaluate its performance without pretraining on the classification dataset (detailed in Section 3.2). This comparison highlights the model's adaptability, demonstrating its ability to rapidly generalize to unseen tasks through a combination of pretraining and fine-tuning. As shown in right panel of Figure 4, we fine-tune the model on datasets of equal size and train for the same number of epochs to ensure a controlled setting. The results indicate that the pretrained

metamodel achieves significantly faster convergence when adapted to new tasks, underscoring its strong generalization and adaptability in continual learning scenarios.

### 5.3 CASE STUDY: ARISING ABILITIES OF MODEL

> **Task: Extract all gene names in the give text**
>
> In the course of Hepatitis A HBs - and HBe - antigen as well as HBc ( IgM and IgG ) - , HBs - and HBe - antibodies can be detected .
>
> - - - - - - - - - - - - - - - - - - - - - - - - - - - - - - - - - -
>
> **Answers:**
> **GPT-4o**:['None']      **Untuned model**:['None']      **Tuned model**:['HBs', 'HBe', 'HBc']

Table 3 reveals the model's strong performance across tasks, with notable patterns emerging. On **Biology** tasks, GPT-4o achieves an average F1 of 0.27—barely matching the metamodel's zero-shot performance—highlighting the low quality of GPT-4o annotations. Intriguingly, fine-tuning the metamodel on these noisy labels still yields substantial gains. Manual analysis attributes **74%** of this improvement to the phenomenon illustrated in 5.3, which we term **arising abilities**.

As shown in 5.3, we define arising abilities as **the model's capacity to correct its outputs even when provided with inaccurate annotation guidance from contemporary LLMs**. Similar phenomena have been observed in prior studies (Shao et al., 2025; Ye et al., 2025), which report that models can self-correct under random or deliberately misleading guidance. These works attribute this capability to the elicitation of the model's extensive pretrained knowledge, aligning with our analytical interpretation. To further validate this hypothesis, we conducted a series of controlled experiments, detailed below.

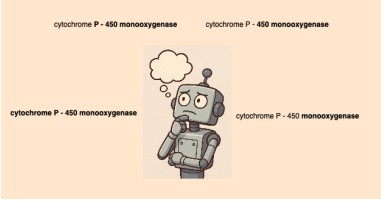

| Models | F1 score |
|---|---|
| Roberta-Large (degraded data) | 0.24 |
| Roberta-Large (original data) | 0.40 |
| Cuckoo (degraded data) | 0.41 |
| Cuckoo (original data) | 0.42 |
| Annotator (GPT-4o) | 0.27 |

(a) Annotated span are marked as Bold      (b) Results of different models on Biology Task

The metamodel's pretraining on IE tasks Peng et al. (2025), which encode entities with positional information, appears to endow it with a strong sensitivity to token structure. We hypothesize that this enables the model to spontaneously extract entities at corresponding positions when faced with new entities sharing similar positional patterns. To test this, we degraded GPT-annotated data by randomizing span start positions while preserving span endings (Figure 5a), retaining primarily positional cues. Fine-tuning on this degraded data yielded performance nearly identical to training on the original annotations, whereas RoBERTa-large suffered a substantial drop (Table 5b). These results suggest that the model's arising ability is driven almost entirely by positional supervision, revealing a striking capability arising from its pretraining knowledge.

## 6 CONCLUSION

This paper proposes a framework for the automated execution of knowledge mining tasks, which decomposes each task into several subtasks and employs a unified model to perform them. Consequently, users only need to provide a task prompt and specify the output format to effortlessly execute a wide range of knowledge mining tasks, while benefiting from performance surpassing that of even the most power modern large language models, as well as 90% inference costs decrease and 20x inference speed increase.

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

## A  CUCKOO FOR TEXT CLASSIFICATION

> **Prompt**
>
> User:
> Choices:
> yes
> no
> **Input Text** Question: Based on above sentence, is the following sentence true or not ?
> This text is about **label**
> Assistant:
> Answer:

We adopt the aforementioned template and leverage the token-level supervision provided by Cuckoo to reformulate the classification task into a more general natural language inference (NLI) problem. An illustrative example is provided below.

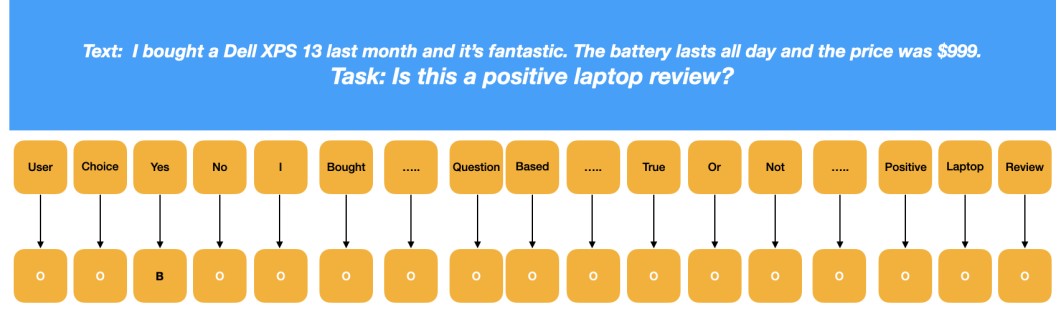

Figure 6: Classification Pretraining

## B  SAMPLE PLANNING CODE

```python
def GPT_pipeline(Input_Corpus):
    labels = ['finance']
    label_results = get_label(Input_Corpus, labels)

    finance_indices = [i for i, result in enumerate(label_results) if
        result[0].lower() == 'yes']
```

```
filtered_texts = [Input_Corpus[i] for i in finance_indices]
if not filtered_texts:
    return []
instruction_spans = "Extract the lecturer of the speak in the given
    text."
spans_results = get_spans(filtered_texts, instruction_spans)
output = []
for idx, orig_idx in enumerate(finance_indices):
    output.append({
        'text': Input_Corpus[orig_idx],
        'spans': spans_results[idx]
    })
return output
```

## C  GENERATING FINE-TUNING SAMPLES

We leverage the metamodel's inherent pretraining knowledge and adopt a heuristic approach to obtain a relatively high-quality fine-tuning dataset. For classification tasks, the generation of fine-tuning samples is illustrated in Algorithm 1, whereas for extraction tasks, we directly employ random sampling.

---

**Algorithm 1:** Classification Training Set Generation

---

**Input:** Corpus $\mathcal{C}$, label $l$, sample size $N$
**Output:** Training set $\mathcal{T}$
Initialize empty set $\mathcal{T}$;
**foreach** *sample $x \in \mathcal{C}$* **do**
$\quad \lfloor$ Compute score $s(x, l)$ using metamodel;
Sort all samples in $\mathcal{C}$ by score $s(x, l)$ in descending order;
Select top $N$ samples $\{x_1^+, \ldots, x_N^+\}$ as positive set $\mathcal{P}$;
Select bottom $N$ samples $\{x_1^-, \ldots, x_N^-\}$ as negative set $\mathcal{N}$;
Construct training set $\mathcal{T} = \mathcal{P} \cup \mathcal{N}$;
**return** $\mathcal{T}$;

---

## D  SAMPLE TASK

> **Sample Task**
>
> 1. retrieve all speaks which is mainly about finance and extract its lecturer
> 2. extract all locations mentioned in the text
> 3. find all talks that address breaking gender stereotypes in modern society, and include all countries mentioned
> 4. retrieve all speaks which is mainly about how mental health influences our daily lives and extract all the institution name mentioned
> 5. retrieve all speaks which is mainly about both health and brain in the speak, then extract their lecturer

## E   HUMAN PROPOSED TASK ON UNLABELED DATASETS

> **Tasks on TED description Dataset**
>
> 1.retrieve all speaks which is mainly about finance and extract its lecturer
> 2.output all speaks which is mainly about mental health and extract its speakers
> 3.return all speaks which is mainly about environment and extract all the locations mentioned in the text
> 4.retrieve talks whose main theme is artificial intelligence and list all professions mentioned
> 5.get all talks that center on medicine and identify all disease mentioned
> 6.collect all speaks which is mainly about finance
> 7.give out all speaks which is mainly about health
> 8.retrieve all speaks which is mainly about education
> 9.gather all speaks which is mainly about technology
> 10.output all speaks which is mainly about politics
> 11.Extract all locations mentioned
> 12.Extract all time mentioned
> 13.Extract all countries mentioned
> 14.Extract all website mentioned
> 15.Extract all person mentioned
> 16.retrieve all speaks which is mainly about how artificial intelligence could affect our lives and its lecturer
> 17.gather talks that mainly discuss climate change and its global impact, and provide all countries mentioned
> 18.retrieve all speaks which is mainly about how mental health influences our daily lives and extract all the institution name mentioned
> 19.find talks that analyze the future of work in an automated world, and return the occupation of the lecturer
> 20.get all talks that address breaking gender stereotypes in modern society, and include the lecturer
> 21.retrieve all texts which is mainly about medicine, and extract what the lecturer will talk about
> 22.retrieve all texts which are mainly about health, and extract all the disease and its associated cause
> 23.find all texts which are mainly about literature, and extract all the awards of [PERSON]
> 24.find all texts which are mainly about science, and extract the profession of [PERSON]
> 25.output all texts which are mainly about history, and extract all the events and the time of the events
> 26.retrieve all speaks which is mainly about both health and brain in the speak, then extract their lecturer
> 27.retrieve all speaks which is mainly about both design and creativity in the speak, then extract all artists mentioned
> 28.retrieve all speaks which is mainly about both medicine and surgery in the speak, then extract all countries mentioned
> 29.retrieve all speaks which is mainly about artificial intelligence and ethics in the speak, then extract all location mentioned
> 30.retrieve all speaks which is mainly about artificial intelligence and machine learning in the speak, then extract the lecturer
> 31.gather all texts which is mainly about finance or artificial intelligence, and extract the lecturer
> 32.get all texts which is mainly about education or biology, and extract all professions
> 33.return all texts which is mainly about philosophy or literature, and extract all person mentioned
> 34.output all speaks which centers on literature or philosophy, then extract all the university affiliation
> 35.retrieve all speaks which centers on music or visual arts, then extract the awards
> 36.retrieve all speaks which is mainly about health but is not about brain, then extract their lecturer

37.retrieve all texts which is mainly about environment but is not about climate change, and extract the locations

38.identify all talks mainly focusing on finance but not mentioning technology, then extract all lecturer name metioned

39.find all speeches mainly about artificial intelligence but without any reference to machine learning, then list all researchers mentioned

40.filter all talks centered on technological innovation but not mentioning blockchain, and extract all numbers mentioned

