# OpenReview forum: "A Tale of LLMs and Induced Small Proxies: Scalable Agents for Knowledge Mining"
_ICLR.cc/2026/Conference — ICLR 2026 Conference Withdrawn Submission_

### Official Review · Reviewer_Y8g4 · 2025-10-16

**Soundness:** 2
**Presentation:** 2
**Contribution:** 2
**Rating:** 2
**Confidence:** 4

**Summary:**

This paper introduces Falconer, a framework for scalable knowledge mining that addresses the trade-off between the high cost of LLMs and the brittleness of traditional IE pipelines. Falconer employs a powerful LLM in two key roles: as a "planner" that decomposes high-level user instructions into a sequence of executable steps, and as an "annotator" that generates supervision data from a small corpus sample. These steps are then executed by a lightweight, instruction-following "proxy" model. The framework elegantly unifies diverse knowledge mining subtasks (classification, extraction) into two atomic primitives, get_label and get_span. The authors create new benchmarks to evaluate this approach and demonstrate that Falconer can match or even exceed the performance of state-of-the-art LLMs like GPT-4o on end-to-end tasks, while reducing inference costs by up to 90% and accelerating throughput by over 20x.

**Strengths:**

- The paper seems to address a critical, real-world bottleneck in applied AI: the prohibitive cost and latency of using flagship LLMs for large-scale data processing.

- The experimental validation may be another major strength. The authors test their system across a wide variety of datasets and task types, using both ground-truth labels and LLM-as-a-judge setups.

**Weaknesses:**

- Limited Generality and Reliance on Heuristics: The framework's primary strength—its efficiency—seems to stem from its highly specialized design, which may limit its broader applicability.
- Specialization for Extractive Tasks: The entire system is built upon two atomic primitives: get_label and get_span. This design makes Falconer a highly optimized solution for extractive and classification-based knowledge mining. However, it inherently restricts its use for other important NLP tasks, particularly generative ones like text summarization, translation, or complex reasoning QA that cannot be decomposed into simple span extractions. The paper should perhaps more explicitly frame Falconer as a specialized framework for its target domain, rather than a general-purpose alternative to LLM agents.
- Generalizability beyond cuckoo: The success of the proxy model, particularly the fascinating "arising abilities" phenomenon, is strongly linked to the Cuckoo model and its specific pre-training paradigm. This raises the question of how generalizable the Falconer framework is. Would the framework be as effective with other open-source small language models (e.g., Phi-3, Gemma, Qwen3-0.6B)?

**Questions:**

Please see weakness

---

### Official Review · Reviewer_1m17 · 2025-10-28

**Soundness:** 2
**Presentation:** 3
**Contribution:** 2
**Rating:** 2
**Confidence:** 4

**Summary:**

The paper presents Falconer, a framework that coordinates multiple agents—including LLMs and lightweight proxy models—for a set of information extraction tasks. Empirical studies are conducted using RoBERTa-large as a baseline on both human-labeled benchmarks and GPT-labeled datasets.

My concerns include:

1. Limited technical novelty.

The proposed framework largely builds upon existing paradigms of multi-agent coordination and LLM-based workflow design (e.g., [1] and [2]). The use of powerful closed-source LLMs (e.g., GPT-4.1) as both a planner (for translating natural-language instructions into executable operations) and as an annotator (for generating labels to fine-tune a lightweight model) has been a common practice in a lot of agent projects.

Moreover, the proxy model Cuckoo is open-sourced, and the technical contribution primarily lies in distilling GPT-4.1 into this small model. While practical, this process does not appear to introduce fundamentally new mechanisms or theoretical insights.

2. Unconvincing empirical studies.

The authors use RoBERTa-large as the only baseline, which is relatively outdated. Given the wide availability of powerful and compact open-source models (e.g., Qwen2.5-0.5B-Instruct), this choice weakens the empirical comparison. Since most text-mining tasks (e.g., NER, RE) have already been effectively addressed by recent LLMs, it is not compelling to demonstrate improvements only over RoBERTa. It is also puzzling that RoBERTa achieves 0 accuracy on some tasks under 64- and 512-sample settings. Was the model fine-tuned on the same data as Falconer? If not, the comparison may be unfair or misleading.

Overall, the work lacks both technical novelty and compelling empirical resutls. The proposed approach appears incremental and primarily implements existing multi-agent workflow concepts without introducing substantial new insights.

[1] Sirui Hong et al., MetaGPT: Meta Programming for A Multi-Agent Collaborative Framework

[2] Jian Guan et al., AMOR: A Recipe for Building Adaptable Modular Knowledge Agents Through Process Feedback

**Strengths:**

a multi-agent implemetation on text mining tasks

**Weaknesses:**

see my comments in Summary

**Questions:**

N/A

---

### Official Review · Reviewer_bP2o · 2025-10-31

**Soundness:** 3
**Presentation:** 3
**Contribution:** 3
**Rating:** 6
**Confidence:** 3

**Summary:**

This paper proposes a framework where LLMs are used as planners and annotators for a knowledge mining task. They introduce two basic operators for classification and entity extraction that are defined as meta operators and train a meta model that can be tuned to perform data mining tasks across various tasks.  As part of the model building, they also build an instruction following benchmark for knowledge mining, which is used for building their meta model.  The proposed method is a performant cost cost-effective alternative to costly LLMs to perform full task vs task-specific models, which do not generalize across various tasks.

**Strengths:**

Approach of utilizing frontier models with ICL examples for planner and building a meta model to do actual data mining is an interesting approach with good cost savings while maintaining the overall performance.
Empirical results showing the effectiveness of the approach are extensive.

**Weaknesses:**

How does the approach compare to any task specific fine tuned models? I don’t see any comments on that? Are the gains significant with respect to those models to adapt this approach in practical settings?

**Questions:**

Please check the weaknesses section.

---

### Official Review · Reviewer_jrA7 · 2025-11-01

**Soundness:** 3
**Presentation:** 2
**Contribution:** 3
**Rating:** 4
**Confidence:** 4

**Summary:**

The paper introduces Falconer, a scalable framework for knowledge mining that fuses the reasoning power of large language models (LLMs) with the efficiency of compact proxy models. In Falconer, LLMs serve as planners—decomposing user instructions into structured subtasks—and annotators, generating high-quality supervision to train lightweight proxies. Central to the framework is the unification of classification and extraction into two atomic operations (get label, get span), allowing a single small instruction-aware model (Cuckoo) to replace multiple specialized components. The authors design new benchmarks to evaluate Falconer’s pipeline planning and end-to-end capability, showing that, with just 5% labeled data, their proxy matches or surpasses state-of-the-art LLMs in accuracy while reducing inference cost by up to 90% and accelerating processing by over 20×. Experiments on both labeled and unlabeled corpora demonstrate strong alignment with human and LLM annotations and highlight the metamodel’s adaptability and continual learning ability. The analysis shows Falconer balances performance, efficiency, and flexibility far better than hand-crafted or purely LLM-based systems, and even demonstrates emergent capabilities, such as self-correction on noisy annotations. In summary, Falconer presents an effective, scalable solution for automated knowledge mining by modularizing task planning and leveraging compact, instruction-following proxies.

**Strengths:**

- Presents a new paradigm that merges LLM planning/annotation with lightweight proxies, unifying extraction and classification under a single, instruction-aware model.
- Goes beyond prior work by automating both supervision (via LLM) and modular pipeline construction.
- Extensive benchmarks comparing to strong baselines (RoBERTa, MetaIE, Cuckoo, state-of-the-art LLMs) across labeled/unlabeled data, sample size, continual learning, efficiency, and diverse task types.
- Clear demonstration of performance, cost, and speed trade-offs.
- Investigates “arising ability” where the proxy model can correct or surpass noisy LLM annotations.
- Well-articulated motivation illustrating limitations of existing pipelines and LLM approaches; easy-to-understand scenario illustrations.
- Achieves substantial reductions in inference cost and deployment overhead.
- Shows feasibility of training on as little as 5% of the data.
- Demonstrates that the proxy can sequentially adapt to diverse tasks with minimal degradation.
- New instruction-following benchmarks and tasks, which could become a community resource.

**Weaknesses:**

- The process for validating LLM-generated supervision, mitigating hallucinations or label errors, and aligning with human annotation is unclear. There is little statistical evidence or qualitative error analysis surrounding annotation reliability.

- Construction, coverage, and difficulty of the new benchmarks are not detailed; it is unclear how well they reflect real-world complexity and industrial use cases.

- The main text lacks deeper discussion of where Falconer or its proxy underperforms (relative to LLMs or modular pipelines), how performance trades off with task complexity, or concrete examples of failure cases.

- No confidence intervals, variance, or significance tests are reported alongside experimental results, which limits the assessment of empirical claims.

- No comprehensive reporting on compute cost, memory/latency figures, or open-source code/benchmarks.

- Needs more explicit differentiation and empirical comparison with closely related recent instruction-IE baselines (e.g., MetaIE, Cuckoo).

**Questions:**

- Could you elaborate more on the design and composition of your new instruction-following benchmarks—specifically task types, annotation processes, and how difficulty was determined? Understanding these aspects is crucial for evaluating the validity and generalizability of your empirical claims.

- Are there specific categories of tasks or scenarios where the unified proxy metamodel underperforms compared to traditional modular or LLM-based approaches? Highlighting limitations will help the community better understand the boundaries of Falconer’s practicality.

- How is annotation quality from LLM “generator” validated—are there measures taken to address potential LLM hallucinations, annotation errors, or domain shift, and how might these influence metamodel outcomes? Since the proxy's learning depends heavily on LLM-generated supervision, understanding annotation reliability and any mitigation steps is necessary for assessing practical effectiveness.

- Please provide complete hyperparameter settings, training/inference resource usage, and dataset splits in the main text, or concisely summarize them with explicit pointers to the appendices.  (To enable reproducibility and fair comparison, readers need direct access to these key experimental details.)

---

### Note · Authors · 2025-11-20

I have read and agree with the venue's withdrawal policy on behalf of myself and my co-authors.